# Slow Continuous Ultrafiltration in Regional Citrate Anticoagulation Performed with a Standard Fluid Infusion Central Venous Catheter in Intensive Care Unit for Fluid Overload in Acute on Chronic Heart Failure: A Case Report

**DOI:** 10.3390/jcm12030988

**Published:** 2023-01-27

**Authors:** Federico Nalesso, Federica Stefanelli, Leda Cattarin, Mariaelena Billo, Maddalena Gnappi, Gabriele Partesano, Martina Cacciapuoti, Luciano Babuin, Lorenzo A. Calò

**Affiliations:** 1Nephrology Unit, Department of Medicine, University of Padua, Via Giustiniani 2, 35128 Padova, Italy; 2Cardiologic Intensive Care Unit, Department of Cardiac, Thoracic, Vascular Sciences and Public Health, University of Padua, Via Giustiniani 2, 35128 Padova, Italy

**Keywords:** SCUF, fluid overload, CVC, ADHF, diuretics

## Abstract

Slow continuous ultrafiltration (SCUF) is an extracorporeal therapy able to reduce fluid overload in chronic or acute heart failure resistant to diuretics. An in-vitro study demonstrated the SCUF feasibility using a standard fluid infusion central venous catheter (CVC). We describe the clinical application of this SCUF in regional citrate anticoagulation (SCUF-RCA) in a patient admitted to the Intensive Care Unit for acute decompensate heart failure with severe systemic fluid overload resistant to diuretics. To avoid risks deriving from a new catheterization, we used a pre-existing multi-lumen CVC for drug administration to provide 10 h of SCUF-RCA with a blood flow of 35 mL/min and 100 mL/h of ultrafiltration with a final weight loss of 1 Kilogram without technical and clinical complications. The patient had a hemodynamics improvement with the diuresis recovery from the previous oliguria after the SCUF-RCA. This clinical case can open the use of the SCUF-RCA in the clinical practice to treat the fluid overload unresponsive to maximal diuretic therapy not exposing the patient to the risks and complications related to the use of SCUF with CVC for dialysis and systemic anticoagulation with heparin. Accordingly, this technique may be useful in the treatment of fluid overload in outpatients.

## 1. Introduction

Slow continuous ultrafiltration (SCUF) by central venous catheter (CVC) for hemodialysis is a simple extracorporeal ultrafiltration aimed to reduce the patient’s fluid overload and control the fluid balance in patients with chronic or acute chronic heart failure unresponsive to medical therapy if started before the development of renal dysfunction [1,2]. If the fluid overload due to heart failure is not treated promptly, Cardio-renal Syndrome (CRS) or acute kidney injury (AKI) can develop and require the use of continuous kidney replacement therapy (CKRT) to treat the fluid overload while assuring the blood purification. The use of CKRT is of particular importance especially in critically ill patients with depressed cardiovascular compliance not adequate to guarantee the use of intermittent treatments with higher weight loss in term of milliliters of plasmatic water removed per hour (mL/h) [3,4,5]. In Intensive Care Unit (ICU) standard central venous catheterizations are routinely placed in critically ill patients with acute decompensated heart failure to provide drugs intravenously. Additionally, CVC placement is required for hemodynamic monitoring, blood sampling and in some selected cases for nutrition. The CVC positioning is a burdensome invasive procedure that can concur to additional morbidity, mortality, and costs due to mechanical, infectious, and thrombotic complications [6]. In the standard clinical practice, patients requiring kidney replacement therapy for AKI, or SCUF for the fluid overload during heart failure, need the insertion of an additional catheter for extracorporeal blood purification therapy further increasing the risk of nosocomial complications and requiring the activation of specific protocols for its maintenance and microbiological surveillance [7,8]. The infection related issue was already underlined in the Cardiorenal Rescue Study in Acute Decompensated Heart Failure (CARRESS-HF) that is a randomized trial involving patients hospitalized for acute decompensated heart failure treated with pharmacologic therapy or ultrafiltration, where a higher percentage of serious adverse events including catheter-related infections were reported in the ultrafiltration group over a period of follow up [9].

In the past the use of peripherally inserted CVC and the smaller-bore catheter has been proposed for patients with cardiac failure and fluid overload to avoid the clinical risk deriving from the supplementary catheterization with standard CVC for hemodialysis required by the standard SCUF or CKRT [10]. In heart failure patient with fluid overload unresponsive to diuretic therapy, an innovative solution could be the use of standard infusion CVC as CVC for extracorporeal circulation to provide the SCUF avoiding the potentially harmful procedure to place the second catheterization with CVC for hemodialysis.

In details, the standard infusion CVC commonly used are multilumen catheters which have more than one internal lumen that allows providing different infusions with separate exit points along the catheter. Since SCUF treatments could be performed with a very low blood flow rate [11], using two different lumens of these CVC would be possible to achieve an adequate blood flow rate for the SCUF while the other lumen(s) can be used for the fluid and medication infusion. As previously reported by Nalesso et Al., it was possible to achieve an adequate blood flow and ultrafiltration through the use of standard infusion CVCs in vitro to obtain the fluid removal for the treatment of patients with fluid overload [12]. In this technique, the regional anticoagulation with sodium citrate was used to minimize the filter and circuit clotting due to the low blood flows obtained through the standard CVCs without exposing the patient to a further clinical risk related to the systemic anticoagulation with heparin [12].

## 2. Materials and Methods

A 54-year-old man was admitted to the Cardiac Intensive Care Unit (CICU) of the University Teaching Hospital of Padua for worsening dyspnea, 12 kg weight gain in the past 7 days. He was diagnosed with acute decompensated heart failure (ADHF) with an ejection fraction (EF) of 24% and severe right ventricle dysfunction. According to his medical history the patient was affected by metabolic syndrome, hypercholesterolemia, severe obesity with a BMI of 45, type 2 diabetes mellitus, history of smoking, severe chronic obstructive pulmonary disease (COPD) with interstitial disease and ischemic cardiomyopathy treated with percutaneous coronary intervention (PCI) in 2020. The patient also had bicuspid aortic valve stenosis submitted to surgical replacement with mechanical valve and ICD implantation in 2014. At the admission in CICU, he had blood pressure (BP) of 70/45 mmHg, heart rate of 123 bpm, a 24-h urine volume of 4 L under furosemide 500 mg per day continuous infusion i.v. and was symptomatic for dyspnea with peripheral oxygen saturation (SpO_2_) of 95% without oxygen support. Physical examination revealed tachycardic heart rhythm, bilateral crackles, diffuse wheezes and pitting edema. Laboratory data showed Acute Kidney Injury (AKI) with urea nitrogen level of 22.9 mmol/L and serum creatinine (s-Cr) of 558 umol/L with a baseline value of 90 umol/L in the past 7 days, serum sodium level of 133 mmol/L and serum potassium level of 4.8 mmol/L. At the admission the inflammation blood tests reported C-reactive protein of 4 mg/L, negative procalcitonine and WBC of 9.280 × 10^9^/L. No marker of AKI was routinely used to assess the risk of AKI evolution [13,14] in this patient although urinary NGAL (Neutrophil Gelatinase-Associated Lipocalin) can be dosed routinely in our hospital.

A continuous infusion of dobutamine was prescribed by cardiologists to increase the BP and furosemide at 1 g per day i.v. in continuous infusion became necessary to treat the progressive oligo-anuria.

Although the patient had non-adherent restricted salt and fluid intake, and diuresis responsive to the furosemide infusion, he was unable to achieve his dry weight because of the ADHF with progressive urine output (UO) reduction despite the diuretic therapy. At the moment of nephrologist consultancy, the UO was less then 0.3 mL/kg/h in the past 6 h.

To reduce the severe systemic fluid overload, associated with the low EF and ADHF, we decided to provide a SCUF treatment considered the diuretic resistance and the progressive reduction in the UO. To reduce the clinical risk deriving from a new catheterization with a CVC for hemodialysis, we used a pre-existing common multi-lumen CVC for drug infusion to provide the SCUF considering the episode of AKI and the oliguria due to the pre-renal clinical component secondary to the ADHF.

According to previous in-vitro study by Nalesso et al. [12], we provided the SCUF in Regional Citrate Anti-Coagulation, using the quad-lumen CVC placed in the right jugular vein (8.5 Fr × 20 cm Quad-lumen catheter kit—Benefis Medical Devices). As previously demonstrated in vitro by Nalesso et al. [12], this type of CVC is suitable to achieve an adequate blood flow to obtain at least an ultrafiltration of 200 mL/h. According to current legislation, the use of infusion CVC, being classified as vascular access device, does not configure the off-label use of this infusion catheter to perform the SCUF as the manufacturer’s user manual. In this patient, the 14 G lumen of the CVC was used as arterial line, and the 16 G as venous line (Figure 1); trisodium citrate 4% (Citrasol 4%, B. Braun Avitum A, Melsungen, Germany) was used as regional anticoagulant in pre-dilution and the Prismaflex monitor (Baxter, IL, USA) was set in SCUF in RCA with HF-20 set (Baxter, IL, USA) that provide a 0.2 m^2^ filter with a maximal ultrafiltration of 1 L/h at a blood flow of 50 mL/min. The blood flow (Qb) was set at 35 mL/min, citratemia at 3.0 mmol/L, calcium compensation at 100% and the weight loss (ultrafiltration, UF) at 100 mL/h (Figure 2). These parameters have determined in the treatment a Filtration Fraction (FF) of 7% with an ionized calcium in the circuit of 0.40 mmol/L (post filter ionized calcium) ensuring the anticoagulation with an FF not critical for filter and circuit clotting. During the treatment the mean pressures in the arterial line was −65 mmHg and in the venous line 140 mmHg with minimal oscillations (±15 mmHg).

## 3. Results

After 10 h of treatment with a weight loss of 1 kg the patient showed an improvement in hemodynamics and UO and therefore the SCUF was stopped by the treating cardiologist treating physician. In details after this extracorporeal treatment there was a progressive improvement in the hemodynamics with a reduction in central venous pressure (CVP) and an increase in blood pressure with an increased diuresis. Probably, the improvement in CVP and BP led to the onset of renal recovery with progressive improvement in renal perfusion and recovery of urine output. From the eighth to the tenth hour of treatment there was an increase in diuresis which reached a value of 200 mL/h, that induced the attending cardiologists to discontinue the SCUF treatment.

There were no changes in total and ionized calcium during and after the treatment. The patient during the following 2 days presented a 150 mL/h diuresis responsive to continuous i.v. infusion of 500 mg per day of furosemide and was discharged from CICU with fluid balance control and the resolution of the fluid overload. On discharge from the CIUC the patient presented a total weight loss of approximately 10 kg compared to the weight gain of 12 kg on CIUC admission. After the resumption of diuresis, the patient had an episode of sepsis of probable abdominal origin due to perforation. The septic state led to the patient’s death even though a therapeutic procedure with specific antibiotics had been started. No autopsy was performed and so the cause of the sepsis is only speculated.

## 4. Discussion

In this patient the SCUF in RCA allowed to reduce the central hyper-hydration decreasing the pre-load of the right heart. The reduction of the CVP determined a progressive improvement of the cardiac function with hemodynamics improvement that induced a valid renal perfusion with recovery of UO and its response to diuretics. Furthermore, the renal recovery was accompanied by the diuresis, validating the hypothesis that the decreased urine output and AKI were caused by a prerenal cause (acute on chronic heart failure and excessive increase in CVP).

This case report demonstrated the feasibility in vivo of SCUF in RCA by standard infusion CVC in patient admitted to ICU in fluid overload resistant to diuretic therapy.

The use of standard infusion CVC for this technique allows to reduce the complications related to the standard CVC for HD placement exposing the patient to a lower risk of CVC-related sepsis. The possibility of using a standard infusion CVC for the SCUF-RCA also allows to reduce the invasive maneuvers in the most fragile patients or in patients with reduced venous assets decreasing the complications related to the placement of the CVC for HD.

Surely the description of this case report opens the perspective to the use of this technique in a larger population of patients, being able to identify any technical and clinical problems not detected in this single case. However, the possibility of performing the SCUF-RCA in patients suffering from heart failure with fluid overload unresponsive to diuretics and carrying standard infusion CVC opens the possibility of considering the use of this technique also in outpatients in which it is possible to place two 14 G cannula needles guaranteeing an adequate blood flow in regional citrate anticoagulation not exposing the patient to the systemic anticoagulation risk. In fact, the diameter of the two lumens used in the CVC during this case report can be replicated by using cannula needles of identical diameters placed in a vein of each upper limb. The possibility of using citrate as a regional anticoagulant in the extracorporeal circulation increases the safety of the SCUF-RCA by not exposing the patient to the risk of circuit and filter clotting due to the low blood flows used, and to the hemorrhagic risk associated with the systemic anticoagulation with heparin as occurs in standard SCUF. Of particular interest is also the safety use of citrate in this technique since the reduced blood flow and the citratemia at 3.0 mmol/L result in a very low risk of citrate accumulation with a citrate load of approximately 6.3 mmol/h, a quantity widely metabolizable in the absence of severe hepatic dysfunction. Surely the execution of this treatment in a single patient does not allow to identify potential CVC malfunctions in the long run and possible malfunctions of the Prismaflex monitor in SCUF-RCA modality with this very low blood flow. To better evaluate these complications it is necessary to perform the treatment in a larger patient population.

## 5. Conclusions

This case report demonstrated the possibility of performing in vivo the SCUF-RCA technique previously tested in vitro using a standard infusion CVC. The feasibility of this treatment in vivo opens the perspective of its use in a larger patient population to assess the clinical and technical complications not highlighted in this single case. The safety and feasibility confirmation of the SCUF-RCA technique by standard infusion CVC will open its use not only in patients admitted to the ICU but also in outpatients in whom cannula needles can be easily positioned at the level of the upper limbs. Furthermore, the use of sodium citrate for circuit anticoagulation is configured as a safe and effective method in preventing filter and circuit clotting without exposing the patient to systemic anticoagulation with hemorrhagic risk.

## Figures and Tables

**Figure 1 jcm-12-00988-f001:**
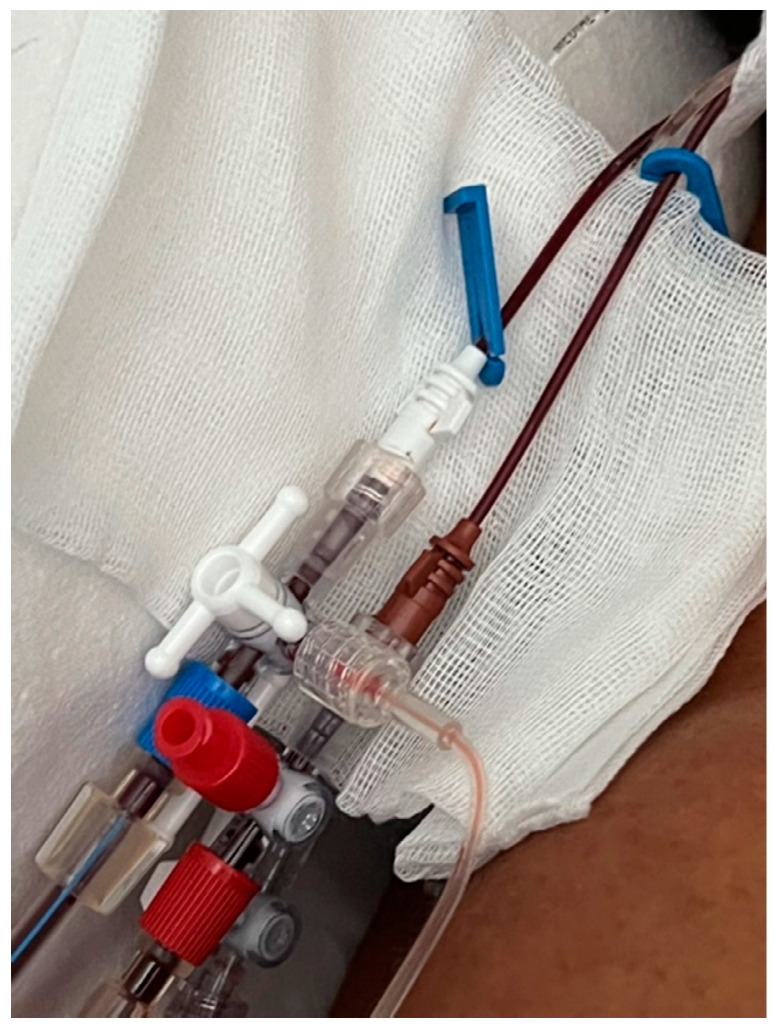
The 14 G lumen of the CVC was used as arterial line, and the 16 G lumen as venous line.

**Figure 2 jcm-12-00988-f002:**
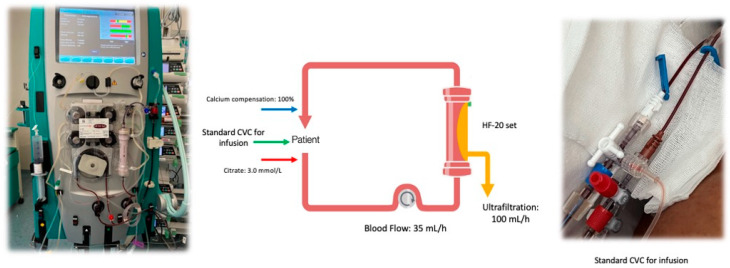
The SCUF with HF-20 set on Prismaflex in regional citrate anticoagulation.

## Data Availability

Not applicable.

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
