# Peer review of "Slow Continuous Ultrafiltration in Regional Citrate Anticoagulation Performed with a Standard Fluid Infusion Central Venous Catheter in Intensive Care Unit for Fluid Overload in Acute on Chronic Heart Failure: A Case Report"

_jcm, 2023, doi:10.3390/jcm12030988_

Round 1

Reviewer 1 Report

The authors present a new application of the infusion catheter (CVC) for SCUF.

Do the authors have a manufacturer's declaration that such a catheter can be used for SCUF? Catheter material may not be suitable for SCUF conditions (low and high pressure)? If there is no such document (the reviewer was unable to find it on the Internet), then the application of such a solution should be approved by the bioethics committee.

What were the pressures achieved in the arterial and venous parts of the system at a flow rate of 35 ml/min?

1kg of ultrafiltration for the entire procedure was probably not crucial in the clinical improvement of the patient, given that the weight gain was 12kg. There were probably other reasons, e.g. increased diuresis after a huge dose of furosemide (500mg) or the control of a septic state?

What CRP or other parameters of inflammation were found in the patient on admission to CICU. Was the patient given antibiotics on admission to CICU?

Line 136 : 200ml/min - or maybe 200ml/h

Author Response

Reviewer 1

The authors present a new application of the infusion catheter (CVC) for SCUF.

Do the authors have a manufacturer's declaration that such a catheter can be used for SCUF? Catheter material may not be suitable for SCUF conditions (low and high pressure)? If there is no such document (the reviewer was unable to find it on the Internet), then the application of such a solution should be approved by the bioethics committee.

The CVC used in this clinical case had previously been tested in vitro to assess the arterial and vein pressures at different blood flows. The CVC is classified as a vascular access medical device. The user manual of the Prismaflex monitor states that the monitor can be used with generic vascular accesses. The CVC used therefore belongs to the vascular access device. The Prismaflex software protects the vascular access from excessively positive or negative pressures. In relation to all these factors, the use of this CVC in this clinical case as a vascular access does not require authorization from the ethics committee as it has been used correctly. The Prismaflex monitor can also be used with standard hemodialysis needles or cannula needles which are also classified as vascular access medical devices.

Nalesso F, Garzotto F, Gobbi L, Cattarin L, Calò LA. In vitro use of standard fluid infusion central venous catheter for slow continuous ultrafiltration feasibility assessment. Artif Organs. 2022 Aug;46(8):1695-1700. doi: 10.1111/aor.14253. Epub 2022 Apr 20. PMID: 35403263.

What were the pressures achieved in the arterial and venous parts of the system at a flow rate of 35 ml/min?

At the blood flow of 35 mL/min the pressure in the arterial line was -65 mmHg and in the venous line 140 mmHg. These pressures were stable for all treatments with minimal variations.

1kg of ultrafiltration for the entire procedure was probably not crucial in the clinical improvement of the patient, given that the weight gain was 12kg. There were probably other reasons, e.g. increased diuresis after a huge dose of furosemide (500mg) or the control of a septic state?

We agree with the reviewer's observation. We emphasize that the total weight gain was 12 kg at the CICU admission and the patient had not achieved a negative balance to reduce this overload. In any case, diuretic therapy prior to the initiation of SCUF had induced a limited weight reduction. At the time of initiation of SCUF treatment, the patient had a fluid overload poorly responsive to diuretic therapy due to increased central venous pressure. The rapid fluid overload reduction of 1 kg by SCUF in association with maximal diuretic therapy allowed a better and rapid recovery of diuresis due to the hemodinamics improvements resulting in the reduction of congestion of the right heart. The septic state occurred following discontinuation of SCUF treatment. . Sepsis temporally did not play a role in the decrease of urine output.

What CRP or other parameters of inflammation were found in the patient on admission to CICU. Was the patient given antibiotics on admission to CICU?

Upon admission to the CICU, the patient did not show any alteration of the inflammation indexes or an increase in white blood cells; no antibiotic therapy had been started.

Line 136: 200ml/min - or maybe 200ml/h

The error identified by the reviewer has been corrected. The correct value is 200 mL/h.

Reviewer 2 Report

This is a very interesting case presentation. However I have few comments

Introduction

1. "with chronic or acute on chronic heart failure" Please rephrase to "with chronic or acute  chronic heart failure"

Materials and Methods

2 "anamnesis " Please replace with "medical history"

3. "500 mg/die " Please explain "die"

4."No marker of AKI was routinely used to assess the risk of AKI evolution in this patient." Please explain, this is not so clear

Results

1."an increase in diuresis which reached a value of 200 mL/min". This value seems wrong

2."He died soon after due to a septic shock ". It seems that apart from the cardiological problem which affected circulation and renal function an infection was also present ? Please make a comment on this

Discussion 

1. Please make a comment on limitations of the case report and of the technique proposed

Author Response

Reviewer 2

Introduction

"with chronic or acute on chronic heart failure" Please rephrase to "with chronic or acute  chronic heart failure"

The sentence has been corrected as requested.

Materials and Methods

"anamnesis " Please replace with "medical history"

The sentence has been corrected as requested.

 "500 mg/die " Please explain "die"

The sentence has been corrected as requested: 500 mg per day.

"No marker of AKI was routinely used to assess the risk of AKI evolution in this patient." Please explain, this is not so clear

In the ICUs of our hospital, it is possible to dose NGAL as marker of renal injury. This test was not used in this patient.

Results

"an increase in diuresis which reached a value of 200 mL/min". This value seems wrong.

The sentence has been corrected as requested. The correct value is 200 mL/h.

"He died soon after due to a septic shock ". It seems that apart from the cardiological problem which affected circulation and renal function an infection was also present? Please make a comment on this

After the diuresis recovery, the patient presented a sepsis of probable abdominal origin due to perforation. The septic state led to the patient's death even though a therapeutic procedure with specific antibiotics had been started. No autopsy was performed and so the cause of the sepsis is only speculated. Sepsis occurred after the recovery of urine output and after the improvement of hemodynamics.

Discussion 

Please make a comment on limitations of the case report and of the technique proposed

Some comments on the limitations of the clinical case and of the technique used have been inserted in the text.

Round 2

Reviewer 1 Report

In the characteristics of the patient, please complete the laboratory data related to inflammation (WBC, CRP, procalcitonin).

If the catheter can be used on-label, it would be beneficial for the potential reader to include a statement about the lack of need to obtain the consent of the bioethics committee and the consent of the patient at the end of the manuscript.

Author Response

Reviewer 1

In the characteristics of the patient, please complete the laboratory data related to inflammation (WBC, CRP, procalcitonin).

The Lab Blood tests required by the reviewer were inserted in the text.

If the catheter can be used on-label, it would be beneficial for the potential reader to include a statement about the lack of need to obtain the consent of the bioethics committee and the consent of the patient at the end of the manuscript.

Informed Consent Statement and Bioethics Committee: we added the text required.

As per the manufacturer's information, the Prismaflex monitor requires a vascular access to perform the extracorporeal circulation. The standard infusion CVC used in this patient is classified as a vascular access medical device thus its use for extracorporeal circulation complies with the requirements of the manufacturer that reports in the user's manual that the most commonly used blood access method for Prismaflex therapies is central venous access and return; dual-lumen venous catheter is the recommended blood access device; however, two single-lumen venous catheters can also be used; the size of the catheter should be adapted to patient and blood flow rate prescription for the extracorporeal therapy. According to these elements Informed Consent and Bioethics Committee have not been requested as they are not necessary due to the on-label use of the CVC.

Reviewer 2 Report

I have no further comments

Author Response

Thanks for the previous comments.